# Pyogenic Spondylodiscitis: Risk Factors for Adverse Clinical Outcome in Routine Clinical Practice

**DOI:** 10.3390/medsci6040096

**Published:** 2018-10-30

**Authors:** John D. Widdrington, Ingrid Emmerson, Milo Cullinan, Manjusha Narayanan, Eleanor Klejnow, Alistair Watson, Edmund L. C. Ong, Matthias L. Schmid, D. Ashley Price, Ulrich Schwab, Christopher J. A. Duncan

**Affiliations:** 1Department of Infection and Tropical Medicine, Royal Victoria Infirmary, Queen Victoria Road, Newcastle upon Tyne NE1 4LP, UK; I.Emmerson@nhs.net (I.E.); milo.cullinan@nuth.nhs.uk (M.C.); eleanor.klejnow@nhs.net (E.K.); ao_watson@hotmail.co.uk (A.W.); edmund.ong@nuth.nhs.uk (E.L.C.O.); matthias.schmid@nuth.nhs.uk (M.L.S.); david.price@nuth.nhs.uk (D.A.P.); uli.schwab@nuth.nhs.uk (U.S.); 2Clinical Research Department, South Tyneside Hospital, Harton Lane, South Tyneside NE34 0PL, UK; 3Department of Microbiology, Royal Victoria Infirmary, Queen Victoria Road, Newcastle upon Tyne NE1 4LP, UK; manjusha.narayanan@nuth.nhs.uk; 4Institute of Cellular Medicine, Inflammation, Immunology and Immunotherapy Theme, Room M3.119, William Leech Building, Newcastle University Medical School, Framlington Place, Newcastle upon Tyne NE2 4HH, UK

**Keywords:** spondylodiscitis, vertebral osteomyelitis, bacterial, antibiotics, outcomes

## Abstract

We aimed to describe the clinical features and outcomes of pyogenic spondylodiscitis and to identify factors associated with an unfavourable clinical outcome (defined as death, permanent disability, spinal instability or persistent pain). In our tertiary centre, 91 cases were identified prospectively and a retrospective descriptive analysis of clinical records was performed prior to binary regression analysis of factors associated with an unfavourable outcome. A median 26 days elapsed from the onset of symptoms to diagnosis and 51% of patients had neurological impairment at presentation. A microbiological diagnosis was reached in 81%, with *Staphylococcus aureus* most commonly isolated. Treatment involved prolonged hospitalisation (median stay 40.5 days), long courses of antibiotics (>6 weeks in 98%) and surgery in 42%. While this was successful in eradicating infection, only 32% of patients had a favourable clinical outcome and six patients (7%) died. Diabetes mellitus, clinical evidence of neurological impairment at presentation, a longer duration of symptoms and radiological evidence of spinal cord or cauda equina compression were independent factors associated with an unfavourable outcome. Our data indicate that spondylodiscitis is associated with significant morbidity and suggest that adverse outcomes may be predicted to an extent by factors present at the time of diagnosis.

## 1. Introduction

Pyogenic spondylodiscitis is defined as a serious infection of the intervertebral disc(s) and/or adjacent vertebrae, and for the purpose of this article incorporates both vertebral osteomyelitis and spondylodiscitis [1]. It may occur due to haematogenous seeding during a bacteraemia, direct spread from an adjacent focus of infection or as a consequence of inoculation during spinal surgery [2]. Although relatively rare, the incidence of spondylodiscitis appears to be increasing (for example in Japan from 5.3 per 100,000 in 2007 to 7.4 per 100,000 in 2010) presumably due to a combination of population aging and advances in diagnostic tools [3,4]. However, there is a critical lack of information about the natural history and long-term clinical outcomes of spondylodiscitis. Furthermore, individual patient and disease-related factors that might contribute to clinical outcome are poorly defined.

The diagnosis of pyogenic spondylodiscitis requires a high index of clinical suspicion combined with early microbiological and radiological investigation (Figure 1). Early diagnosis is important as spondylodiscitis is often associated with abscess formation in the epidural space and adjacent soft tissues and muscle. The effects of local inflammation and abscess formation can lead to spinal cord compression and/or vertebral column instability, resulting in permanent neurological deficits. Management involves a combination of prolonged antimicrobial therapy and surgical intervention but is based on a very limited evidence base [1,4].

To address the clinical need for better data on natural history and outcome, we sought to describe the presentation, management and outcome of pyogenic spondylodiscitis in a tertiary centre for spinal surgery and infectious diseases.

## 2. Materials and Methods

Data were collected as part of routine clinical service evaluation at Newcastle Upon Tyne Hospitals NHS Trust, a large teaching trust with 1800 beds that manages over 1.72 million patient contacts per year and is a tertiary referral centre for both Spinal Surgery and Infectious Diseases services. Cases of bacterial spondylodiscitis between 1 May 2013 and 31 May 2016 were prospectively identified upon referral for infectious diseases consultation. The inclusion criteria were: adult patients (aged 16 years or over), a radiological diagnosis of pyogenic spondylodiscitis and a bacterial aetiology confirmed by microbiological culture or a bacterial aetiology not confirmed but thought likely by the treating infectious diseases team. The exclusion criteria were: children (aged under 16 years), confirmed tuberculous (*n* = 2) or fungal (*n* = 1) spondylodiscitis or the absence of clinical outcome data (*n* = 7). Subsequent to the prospective case identification, anonymised details of clinical and radiological presenting features, microbiological diagnosis, and surgical and antimicrobial treatment were retrospectively obtained from clinical and laboratory records and recorded using a standardised proforma and/or directly entered onto a database (Microsoft Excel, Redmond, WA, USA). The degree of neurological impairment was retrospectively determined using the Frankel grading scale of neurological injury [5] using data collected at the time of clinical assessment. The Frankel scale has the following grades differentiated by the residual function below the level of injury: A, complete loss of motor and sensory function; B, preserved sensation only; C, preserved motor activity of no practical use to the patient; D, preserved impaired but useful motor function; and E, normal motor and sensory function. This functional scale has been previously used in studies of spondylodiscitis [6,7]. Outcome data were collected retrospectively using clinical and electronic patient records. Patients were classified as having a favourable outcome (complete recovery) or an unfavourable outcome, defined as either death during hospital admission or the presence of physical disability and/or persistent pain at one year or, if earlier, the point of discharge from follow-up.

Statistical analysis was carried out using GraphPad Prism 6 (GraphPad, La Jolla, CA, USA) software using *t*-test for continuous data and Fisher’s exact test for categorical data. Multivariate binary logistic regression analysis of factors that were significant in univariate analysis was carried out using Minitab 17 (Minitab Limited, Coventry, UK). Two-tailed α < 0.05 was considered significant.

## 3. Results

### 3.1. Baseline Features

We identified 98 patients with pyogenic spondylodiscitis. Full outcome data were available on 91/98 patients. Patients in the cohort had a median age of 61 years (range 17–90 years). Fifty-five (60%) of these patients were male, which is in keeping with the known male preponderance. A pre-existing condition causing impaired immunity was present in 31 (34%) patients: 16 had diabetes mellitus, 11 were on various immunosuppressive agents for autoimmune diseases, six had active cancer and one had a splenectomy (three patients had more than one of these conditions). In addition, 20 (22%) patients had a history of spinal surgery within the past three years: six infections occurred after decompressive laminectomy, six after discectomy, five after spinal fixation, one after vertebroplasty, one after insertion of a spinal stimulator and one after resection of a retroperitoneal sarcoma. Four patients (4%) were active injecting drug users.

### 3.2. Clinical Findings

Although the majority of patients with spondylodiscitis reported spinal pain and/or radiculopathy at presentation, other presenting symptoms were less universal (Figure 2A). Notably, self-reported or objectively documented fever was absent in 34 (37%) patients. At presentation, 46 patients (51%) had a Frankel grading score of A–D indicating various degrees of impairment of motor function; of these patients, 31 had a spinal epidural abscess, six had vertebral instability, five had a prevertebral or paravertebral abscess and in the remaining four spondylodiscitis was the only identifiable cause for the neurological impairment, which was presumably secondary to nerve root impingement. There was considerable variation in the reported duration of symptoms, but spondylodiscitis generally had a subacute presentation with median symptom duration of 26 days (range 1–203 days) prior to diagnosis.

The diagnosis of spondylodiscitis was suggested by characteristic features on magnetic resonance imaging (MRI) of the spine. Spondylodiscitis occurred throughout the spine, with a preponderance for the lumbar and lumbosacral spine (50 patients, 55%), and more than one disc space was involved in 27 (30%) patients (Figure 2B). Furthermore, 55 (60%) patients had spinal epidural abscess formation and 32 (35%) had radiological evidence of spinal cord/cauda equina compression at presentation (Figure 2C).

The most common source of infection was haematogenous spread (48 patients, 53%). Twenty patients (22%) had an infection following spinal surgery while seven patients (8%) had spread from an identified contiguous source of infection (e.g., intra-abdominal infection). The underlying source of infection was not identified in 16 patients (17%) (Figure 2D).

### 3.3. Microbiology

A microbiological aetiology was confirmed in 74 (81%) patients, 62 of whom had a single organism identified and the remaining 12 had multiple organisms identified. Bacteria were isolated from blood cultures in 40 patients (51% of the 78 patients from whom blood cultures were obtained), image-guided biopsy samples in 21 patients (72% of 29 patients) and surgical samples in 25 patients (66% of 38 patients). Gram-positive bacteria were most commonly cultured, followed by Gram-negative bacteria and anaerobes. The most commonly identified bacterial pathogens were *Staphylococcus aureus* (35 patients, 38%), coagulase negative Staphylococci (11 patients, 12%) and *Escherichia coli* (11 patients, 12%) (Table 1). Although antimicrobial resistance data are not included here, this was not a major problem in our cohort of patients, as illustrated by the finding that only 1/35 (3%) of patients in which *S. aureus* was isolated had a methicillin-resistant strain. As expected, there was a significant association between the isolation of coagulase negative Staphylococci and spondylodiscitis post-spinal surgery (5/20 patients with post-surgical infection vs. 6/69 other patients, *p* = 0.049). There were no other associations between baseline factors (e.g., surgical infection, immunocompromise, diabetes, etc.) and microbiological diagnosis.

### 3.4. Treatment

Surgical intervention occurred in 38 patients (42%), 13 of whom required more than one procedure (Figure 3A). A decompressive laminectomy was the most common surgical procedure, while 16 patients (18%) required spinal fixation or fusion (either at the time of presentation (14/16) or following antimicrobial treatment (2/16)). Epidural abscess was drained surgically in 28 (51%) of the 55 patients with radiological evidence of epidural abscess (Figure 3B).

In general, patients were treated with systemic antibiotic therapy for a minimum of 6 weeks (89 patients, 98%). In most cases, this consisted of an initial intravenous antibiotic-containing regimen (88 patients, 97%) for a minimum of 4 weeks (80 of these 88 patients, 91%) followed by an additional oral regimen (70 of these 88 patients, 80%). Combination therapy with two antibiotics was most common (Figure 4A). Beyond these broad principles, there was wide variation in the actual antibiotic regimens used for both intravenous and oral therapy and the total duration of treatment (Figure 4B). The same antibiotic regimen was used throughout treatment in the minority of patients (seven patients, 8%), with most having two or three different regimens (Figure 4C).

### 3.5. Clinical Outcomes

Spondylodiscitis was generally associated with a long duration of hospital admission with a median stay of 40.5 days (range 6–207). The majority of patients had an improvement in symptoms and motor/sensory function after treatment, as reflected by changes in the Frankel grading score (Figure 5A). However, in addition to a fatal outcome in six (7%) patients, there was significant long-term morbidity in the majority with only 29 (32%) patients experiencing a complete recovery. In the other patients, 28 (31%) had significant ongoing spinal pain or radiculopathy requiring regular analgesia, 24 (26%) had persistent neurological impairment and four (4%) had vertebral instability requiring further surgical or orthotic intervention following treatment of the infection (Figure 5B). Subsequent analysis revealed that the following factors at the time of diagnosis of spondylodiscitis were significantly predictive of unfavourable outcome: diabetes mellitus, clinical evidence of neurological impairment at presentation (Frankel grading score A–D), a longer duration of symptoms prior to presentation and radiological findings suggestive of spinal cord/cauda equina compression (Table 2 and Table 3).

## 4. Discussion

This study provides a comprehensive assessment of the presentation, diagnosis, management and outcomes of 91 prospectively-identified adults treated for bacterial spondylodiscitis over three years in a tertiary referral centre. In keeping with previous studies, we have shown that spondylodiscitis is more common with increasing age and male gender and that the presenting features are highly variable [2,8,9,10,11,12]. In this “real world” dataset, a combination of blood and biopsy culture enabled a microbiological diagnosis to be reached in the majority of patients. Indeed, half of patients had a positive blood culture, reinforcing the importance of this simple investigation in patients with suspected spondylodiscitis. *S. aureus* was the most common pathogen, consistent with existing reports [2,11]. Importantly, our data demonstrate that, despite prolonged and microbiologically-targeted antibiotic therapy, and a reasonably high rate of surgical intervention, spondylodiscitis was associated with significant adverse outcomes including prolonged hospital stay, persistent pain, long-term disability and death, with less than one third of patients making a complete recovery. As we identified no evidence of infection relapse, these sequelae are most likely to relate to the damage done during the initial illness.

Few studies report on patient-related factors which are associated with poor outcome in spondylodiscitis. We found that an unfavourable outcome was predicted by the presence of the following risk factors at the point of diagnosis: diabetes mellitus, longer interval between symptom onset and diagnosis, neurological impairment at diagnosis (Frankel grade A–D) and radiological evidence of spinal cord or cauda equina compression. In agreement with our findings, a prospective study of 81 patients revealed that the presence of a neurological deficit at diagnosis and delay in presentation of greater than 60 days was significantly associated with persistence of neurological deficit or relapse of infection [9]. Similarly, in a retrospective analysis of 253 patients, death or incomplete recovery from spondylodiscitis was associated with neurological compromise at diagnosis, delayed diagnosis and nosocomial acquisition of infection [10]. Other studies have focused on specific outcomes, for example severe neurological dysfunction or death. A large retrospective study which specifically addressed risk factors for mortality (but not morbidity) identified increased age, diabetes mellitus, cirrhosis, malignancy, haemodialysis use and concomitant infective endocarditis as independent factors [3]. Another retrospective study of 62 patients found that an adverse neurological outcome was significantly more likely in those patients with concomitant spondylodiscitis and infective endocarditis. It should be noted that in our study very few patients were diagnosed with concomitant endocarditis, although only 55 patients (60%) had indications for an echocardiogram [12] More recently, a study incorporating patients recruited to an open-label randomised controlled trial (RCT), discussed further below, examined factors that were associated with severe neurological deficit at baseline (but not outcome): these factors were cervical or thoracic spinal involvement, *S. aureus* infection and C-reactive protein > 150 mg/L [13].

Collectively, the outcome from spondylodiscitis appears to be largely governed by factors present at the time of diagnosis. For several reasons, spondylodiscitis is a difficult diagnosis to make without a high index of suspicion, considering its rarity compared to more common causes of back pain. Therefore, diagnostic delay is a common theme in the literature [4,10,14]. Timely diagnosis appears to be vital to prevent the onset of neurological deficit and/or spinal instability [1] and could be aided by increased awareness of the diagnosis amongst primary care and general physicians as well as the wider public, and potentially by expanding access to spinal imaging. Reducing diagnostic delay is critical, since any advances in treatment are unlikely to yield significant benefits unless allied to improvements in diagnosis. While the presence of factors associated with a worse outcome cannot necessarily be mitigated, our study highlights that there might be a potential role for individualised therapy (e.g., intensified antibiotics or adjunctive therapy) to improve outcome in patients with pre-existing diabetes and baseline neurological impairment.

While there was considerable heterogeneity in terms of antimicrobial therapy used in our dataset, the majority of patients received a minimum duration of 6 weeks of therapy, generally involving combination antibiotic therapy, with an initial intravenous antibiotic-containing regimen followed by an additional oral antibiotic regimen. This approach is in accordance with that reported elsewhere and international consensus guidelines suggesting that a minimum of 6 weeks of antibiotic therapy is required to treat spondylodiscitis [1,15]. However, such guidelines are predominantly based on expert opinion and the underlying evidence is of poor quality [11,16].

Thus, the optimal duration of antibiotic therapy for spondylodiscitis remains uncertain. The best evidence available is from an open label non-inferiority RCT of 359 patients which showed no difference in clinical cure or adverse events with 6 weeks compared to 12 weeks of antibiotic therapy [17]. This is consistent with a previous retrospective study that found no difference in the risk of mortality or relapse in 36 patients treated for less than 6 weeks compared to 84 treated for longer [18], although clearly allocation bias cannot be excluded in this study design. There is very limited evidence to determine whether intravenous or high-bioavailability oral therapy is superior, although the intravenous route may be necessary in certain specific scenarios (e.g., the absence of an effective oral option or inability to tolerate oral antibiotics) [16]. While the previously mentioned randomised controlled trial was not designed to assess this, 52% of those enrolled received less than 14 days intravenous therapy and there was no increase in treatment failure in those having intravenous therapy for less than 7 days [17]. However, caution is needed when interpreting such post-hoc analyses. A Cochrane review of antibiotic therapy for osteomyelitis (predominantly post-traumatic osteomyelitis of the lower limb) concluded that the route of administration of antibiotics does not appear to alter the effectiveness of treatment [16], although the quality of studies included was low and it is not clear that these data can be extrapolated to spondylodiscitis. Finally, while many physicians routinely prescribe combination antibiotic therapy, the efficacy of this approach has not been formally assessed in spondylodiscitis. The observation that combination therapy with rifampicin is associated with better outcomes in osteomyelitis associated with prosthetic material may be relevant to at least a subset of patients [4,19]. Collectively, the uncertainties summarised above point to the need for randomised controlled trials to determine the optimal antibiotic management of spondylodiscitis. For obvious reasons, when considering patient allocation to these studies it will be important to ensure that the baseline risk factors we have identified are equivalently distributed between intervention and control groups.

Another area of some uncertainty concerns the role for surgical intervention in the treatment of spondylodiscitis. Surgery typically involves decompression of the spinal cord and debridement of infected tissues and/or drainage of abscesses [11]. Indications include the presence of neurological injury/spinal cord compression, vertebral instability, failure of medical therapy and epidural or paraspinal abscesses [3,4]. A retrospective study of 90 patients in Taiwan found that the 43 patients having early surgical intervention had a shorter hospital stay and reduced spinal deformity, pain and disability compared to those treated conservatively with antibiotics alone [20]. However, conservative management is increasingly practiced, even in selected patients with spinal epidural abscess [8,21,22], and this was also the case in our cohort. To some extent, this may be due to concerns about implanting spinal metalwork into an area of active infection. This concern was not supported by two recent small retrospective studies that did not demonstrate an increased risk of infection relapse [23,24], although clearly more data are needed.

We acknowledge some limitations of our study. Whilst cases were identified prospectively, outcome data were collected and analysed retrospectively and therefore may be subject to bias. Our study population represents patients who were managed jointly by infectious diseases and spinal surgical services at a single institution and might reflect the more severe end of the disease spectrum. Whether our findings (for example, the low rate of infection relapse) would be generalizable to patients managed without infectious diseases input remains unclear. Furthermore, although this is the largest UK cohort and the sample size compares well to previous studies, the sample size may have limited our power to identify additional treatment-related factors important to clinical outcome, which will be better assessed through a large multi-centre observational study. Finally, it can also be argued that post-surgical infections may represent a distinct pathogenic process to spondylodiscitis from other causes, particularly in view of the differences in microbiological aetiology reported here. However, including post-surgical cases reflects the reality that the treatment principles are similar and is consistent with the approach taken in other cases series and the randomised controlled trial by Bernard et al. [9,10,17].

## 5. Conclusions

Spondylodiscitis is a rare but serious condition that leads to significant long-term morbidity. In this study, we have identified several important findings that contribute to our understanding of this neglected disease. These include the observations that: (a) there is often a substantial interval between symptom onset and diagnosis; (b) long-term sequelae such as pain are common despite adequate treatment of infection; and (c) several easily identifiable baseline factors (including duration of symptoms, diabetes mellitus and neurological involvement at presentation) appear to predict adverse clinical outcome. Our findings highlight the need for further research, particularly to assess the impact of interventions to reduce diagnostic delay and to optimise antibiotic and/or surgical management. Such research has considerable potential to improve the outcome for patients with spinal infections.

## Figures and Tables

**Figure 1 medsci-06-00096-f001:**
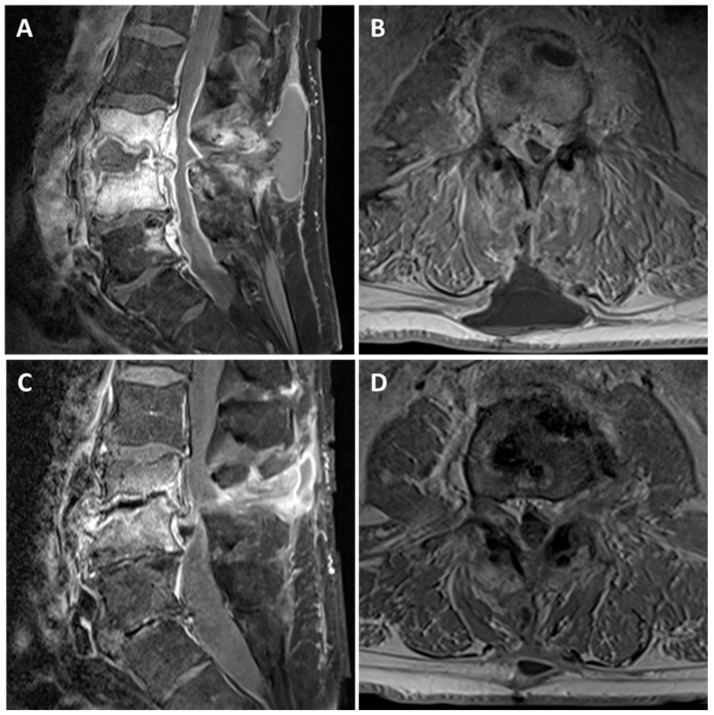
The appearances of L3/4 spondylodiscitis before and after antimicrobial therapy. Representative T1 fat saturated post-gadolinium sagittal (**A**,**C**) and axial (**B**,**D**) Magnetic Resonance images. (**A**,**B**) At presentation, there is a rim enhancing collection within L3/4 disc space with destruction of the adjacent vertebral body end plates. In addition, there is an associated enhancing epidural phlegmon causing severe cauda equina compression and a homogenous subcutaneous fluid collection. (**C**,**D**) Six months later, following antimicrobial therapy, there is significant improvement in the inflammatory appearances at L3/4 disc space and the fluid collections. However, there is a significant loss of disc height at L3/4 leading to moderate-severe canal stenosis.

**Figure 2 medsci-06-00096-f002:**
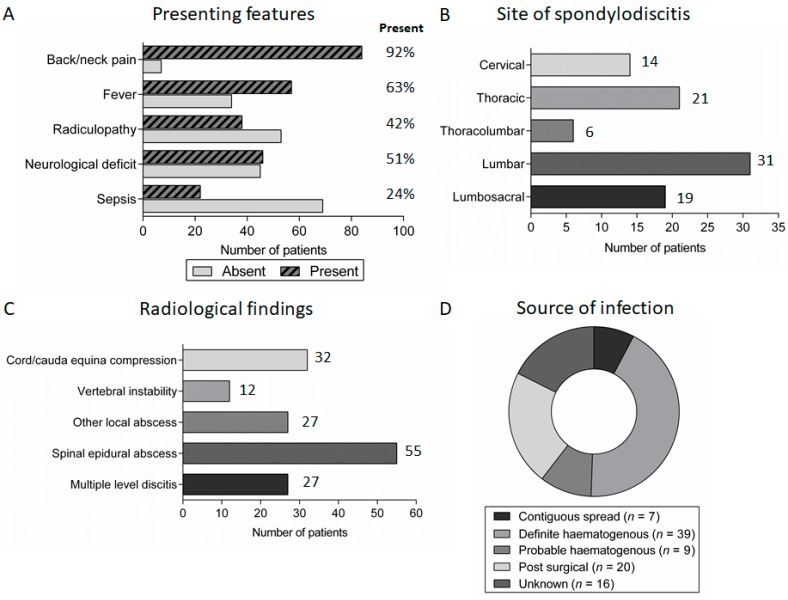
Symptoms, radiological features and underlying source of spondylodiscitis: (**A**) the presence/absence of important symptoms in patients presenting with discitis; (**B**) the distribution of spondylodiscitis between different anatomical regions; (**C**) the number of patients with complications of spondylodiscitis identified on MRI imaging at presentation; and (**D**) the underlying source of infection in patients with spondylodiscitis.

**Figure 3 medsci-06-00096-f003:**
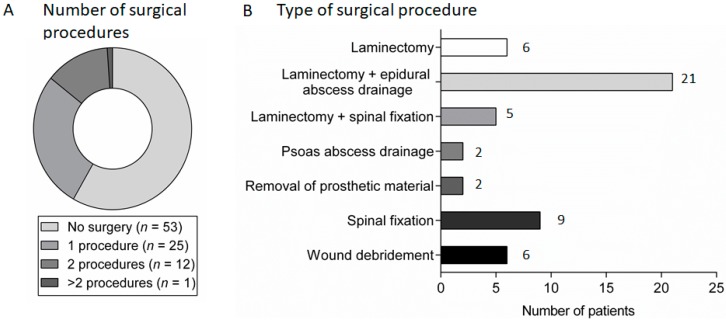
Surgical management in patients with spondylodiscitis: (**A**) the number of surgical procedures carried out in patients with spondylodiscitis; and (**B**) the frequency of different types of surgical procedures carried out in patients with spondylodiscitis.

**Figure 4 medsci-06-00096-f004:**
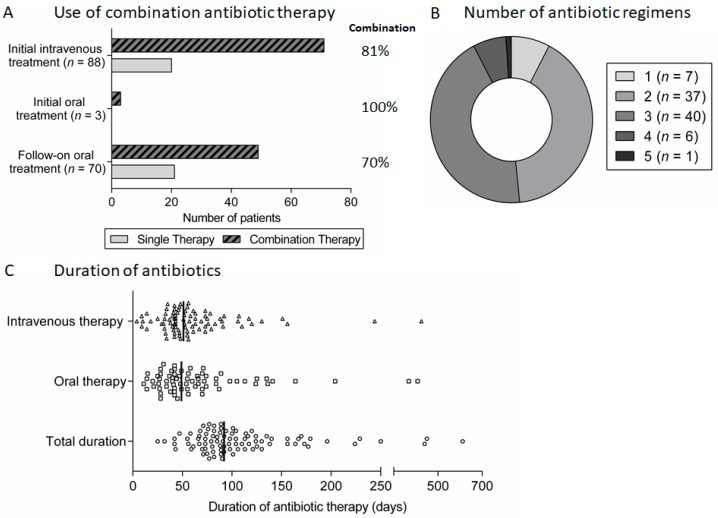
Antibiotic therapy for spondylodiscitis: (**A**) proportion of patients treated using combination antibiotic regimens during initial intravenous or oral therapy and follow-on oral therapy after initial intravenous therapy; (**B**) the number of different antibiotic therapy regimens used in patients treated for spondylodiscitis; and (**C**) scatter plot of the duration of intravenous, oral and total antibiotic therapy for spondylodiscitis, with lines indicating the median duration.

**Figure 5 medsci-06-00096-f005:**
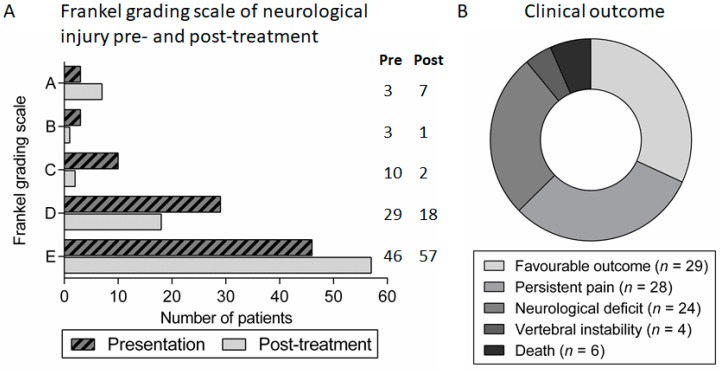
Clinical outcomes in patients with spondylodiscitis: (**A**) Frankel grading scale of neurological injury in patients with spondylodiscitis at presentation and at follow-up review after treatment; and (**B**) proportion of patients with different clinical outcomes following treatment for spondylodiscitis.

**Table 1 medsci-06-00096-t001:** Bacteria isolated from patients with spondylodiscitis.

Group	Pathogen	Total	Single Organism Isolated	Multiple Organisms Isolated
**Gram positive**	All	58	46	12
	*Staphylococcus aureus*	35	33	2
	Coagulase negative Staphylococci	11	6	5
	*Enterococcus faecalis*	3	1	2
	*Streptococcus pneumoniae*	3	3	0
	Viridans group Streptococci	3	1	2
	Group B Streptococci	2	2	0
	Group C/G Streptococci	1	0	1
**Gram negative**	All	17	13	4
	*Escherichia coli*	11	10	1
	*Klebsiella pneumoniae*	2	1	1
	Others	4	2	2
**Anaerobes**	All	9	3	6
	*Proprionibacterium acnes*	4	2	2
	Others	5	1	4

The causative organisms isolated by microbiological culture from blood, biopsy and surgical samples are listed with subgrouping depending on whether they were isolated alone or with other organisms.

**Table 2 medsci-06-00096-t002:** Factors predicting outcome from spondylodiscitis.

Risk Factor	Total (*n* = 91)	Favourable Outcome (*n* = 29)	Unfavourable Outcome ^1^ (*n* = 62)	*p*-Value ^2^ Univariate Analysis
Background				
Age (median (range))	62.5 (17–91)	59 (17–90)	64 (35–91)	0.121
Diabetes mellitus (*n* (%))	16 (18)	1 (3)	15 (24)	0.017 *
Immune compromise (*n* (%))	18 (20)	3 (10)	15 (24)	0.162
Post-surgical (*n* (%))	16 (18)	5 (17)	11 (18)	>0.99
Presenting features				
Sepsis (*n* (%))	22 (24)	7 (24)	15 (24)	>0.99
Frankel grading scale A–D (*n* (%))	46 (51)	5 (17)	39 (63)	<0.001 *
Duration of symptoms (median days (range))	36 (1–203)	27 (1–71)	41 (2–203)	0.049 *
C-reactive protein (median mg/L (range))	170 (4–508)	143 (15–395)	181 (4–508)	0.157
White blood cell count (median × 10^9^ (range))	13.1 (6.0–33.7)	12.2 (6.6–24.3)	13.6 (6.0–33.7)	0.18
Endocarditis	4 (4)	1 (3)	3 (5)	>0.99
Radiological features				
Epidural abscess (*n* (%))	55 (60)	16 (55)	39 (63)	0.467
Multiple level discitis (*n* (%))	27 (30)	6 (20)	21 (34)	0.228
Cord/cauda equina compression (*n* (%))	32 (35)	5 (17.2)	27 (44)	0.018 *
Vertebral instability (*n* (%))	12 (13)	2 (7)	10 (16)	0.325
Microbiological diagnosis				
Bacteraemia (*n* (%))	47 (52)	18 (62)	29 (47)	0.187
*Staphylococcus aureus* (*n* (%))	35 (39)	10 (35)	25 (40)	0.649

^1^ Unfavourable outcome was defined as death, long-term physical disability or persistent pain requiring regular analgesia. ^2^ The significance of the differences between the groups with favourable and unfavourable outcomes was analysed using independent t-tests for continuous variables and Fisher’s exact testing for categorical variables. * indicates *p* < 0.05.

**Table 3 medsci-06-00096-t003:** Binary logistic regression of factors predicting an unfavourable outcome from spondylodiscitis.

Risk Factor	Odds Ratio	95% Confidence Interval	*p*-Value
Diabetes mellitus	5.88	1.40–24.68	0.008
Frankel grading scale A–D	4.52	1.25–16.39	0.019
Duration of symptoms	1.02	1.00–1.04	0.019
Spinal cord/cauda equina compression	6.49	1.42–29.70	0.009

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
