# Peer review of "Pyogenic Spondylodiscitis: Risk Factors for Adverse Clinical Outcome in Routine Clinical Practice"

_medsci, 2018, doi:10.3390/medsci6040096_

Round 1
Reviewer 1 Report
Thank you very much to give me the opportunity to read this interesting study about pyogenic spondylodiscitis in adults, that aims to describe clinical features, and to describe primary risk factors for adverse clinical outcome.
This present study presents a large prospective consecutive case series (including all eligible patients identified by the researchers during the study registration period) that reported about pyogenic spondylodiscitis in adults.
It provides significant data about this affliction in terms of epidemiology, clinical and biological presentations and microbacteriological aetiology. In their study, the authors enrolled 91 consecutive adult patients with pyogenic spondylodiscitis and demonstrated that S. aureus was the pathogen most commnly isolated. Authors also suggested that risk factors such as diabetes mellitus, duration of symptoms, neurological troubles and radiological evience of spinal cord compression were associated with unfavourable outcome.
The study is well done, the results are interesting but I have a few comments and any suggestions to do to the authors.
In the introduction you speak about vertebral osteomyelitis and spondylitis. Which difference are you making unbetween the two terms?
You also speak about discitis. Garron et al. suggested that pure discitis as reported by some authors does not exist for vascular anatomical reasons. In young children, the metaphysis of the vertebral body exhibits a rich vascular ring, which creates an anastomosis with the adjacent metaphyseal vascular ring through several branches adjacent to the posterolateral region of the disk. The vascular supply of the disc in children is made up of vessels that come across the cartilaginous vertebral plate and into the disc ring. After 8 years of age, these vessels disappear, but a dense anastomotic network of vessels remains in the postero-lateral region of the disc. Thus, some authors consider that all bacterial infections are primarily located in the metaphyseal region of the vertebral body, with the microorganism first crossing the cartilaginous vertebral plate, running through the surface of the disc via the anastomotic branches, infecting the adjacent vertebral metaphysis and finally reaching the disc space between the two vertebral bodies involved. What is opinion about this physiopathological explanation?
Inclusion and exclusion criteria have to be better explained.
In the series, the authors collected 20 patients that presented surgical site infections following spinal surgery. I do not think that these cases may be considered as pyogenic spondylodiscitis, but they are secondary spinal infections. To my opinion, surgical site infections following spinal surgery are affections, with different microbiological reality, treatment and outcome.
If you consider that these cases of surgical site infections following spinal surgery have to be collected in your series, it will be important to know what surgeries were realized (discectomies, spondylodesis, radicular neurolysis for radiculopathies, decompressive laminectomies,…?).
More than 50% of their patients had impairment of motor function. Could authors explained the real cause of neurological impairment (compression due to epidural abscess, instability, narrow canal,…?).
Could the authors explain which were their indications for performing drainage of epidural abscesses?
In the discussion, the authors evoke the severity of illness index; could they explain index illness to the readers?
Author Response
Department of Infection and Tropical Medicine,
Royal Victoria Infirmary,
Queen Victoria Road, Newcastle upon Tyne,
NE1 4LP,
United Kingdom
17th October 2018
Re: medsci-363764 - Pyogenic spondylodiscitis: risk factors for adverse clinical outcome
Dear Reviewer
Thank you for your email of 10th October 2018 giving us the opportunity to respond to the comments made following the review of our article. We have responded to each of these comments below and made requested changes to the manuscript. We believe the manuscript is improved as a result of these changes and hope that it will now be acceptable for publication in Medical Sciences.
In the text below we have listed the comments in the order that they appeared in each review email and numbered them C1, C2 etc. with our corresponding responses beneath the relevant comment and numbered R1, R2 etc. As requested we have produced a revised manuscript in which our changes are clearly marked. The line numbers included in the responses corresponds to those in the ‘Simple Markup’ tracked version of the Word Document.
We would be happy to provide any further information that you require.
Yours faithfully
Dr John Widdrington, on behalf of the authors
Open Review
(x) I would not like to sign my review report
( ) I would like to sign my review report
English language and style
( ) Extensive editing of English language and style required
( ) Moderate English changes required
( ) English language and style are fine/minor spell check required
(x) I don't feel qualified to judge about the English language and style
Yes | Can be improved | Must be improved | Not applicable | |
Does the introduction provide sufficient background and include all relevant references? | (x) | ( ) | ( ) | ( ) |
Is the research design appropriate? | ( ) | (x) | ( ) | ( ) |
Are the methods adequately described? | ( ) | (x) | ( ) | ( ) |
Are the results clearly presented? | (x) | ( ) | ( ) | ( ) |
Are the conclusions supported by the results? | ( ) | (x) | ( ) | ( ) |
Comments and Suggestions for Authors
Thank you very much to give me the opportunity to read this interesting study about pyogenic spondylodiscitis in adults, that aims to describe clinical features, and to describe primary risk factors for adverse clinical outcome.
This present study presents a large prospective consecutive case series (including all eligible patients identified by the researchers during the study registration period) that reported about pyogenic spondylodiscitis in adults.
It provides significant data about this affliction in terms of epidemiology, clinical and biological presentations and microbacteriological aetiology. In their study, the authors enrolled 91 consecutive adult patients with pyogenic spondylodiscitis and demonstrated that S. aureus was the pathogen most commonly isolated. Authors also suggested that risk factors such as diabetes mellitus, duration of symptoms, neurological troubles and radiological evidence of spinal cord compression were associated with unfavourable outcome.
The study is well done, the results are interesting but I have a few comments and any suggestions to do to the authors.
Response: We are grateful to the reviewer for these supportive comments about this manuscript.
C1: ‘In the introduction you speak about vertebral osteomyelitis and spondylitis. Which difference are you making unbetween the two terms?’
R1: There is considerable variation in the terms used to describe spinal infection. These can have imprecise and overlapping definitions and we regret any confusion that this lack of clarity might introduce. It is our view that the terms ‘vertebral osteomyelitis’ and ‘infectious spondylitis’ are often used interchangeably to describe the same clinical entity. Rather than an attempt to differentiate between these terms (which in practice are typically concurrent), the first sentence of our introduction was intended to indicate that we were using the term ‘pyogenic spondylodiscitis’ throughout our manuscript as an umbrella term to describe cases of spinal infection involving the vertebrae and/or the intervertebral discs. To reduce any confusion, we made the following statement has been added:
Lines 38-40: ‘Pyogenic spondylodiscitis is defined as a serious infection of the intervertebral disc(s) and/or adjacent vertebrae, and for the purpose of this article incorporates both vertebral osteomyelitis and spondylodiscitis.’
C2: ‘You also speak about discitis. Garron et al. suggested that pure discitis as reported by some authors does not exist for vascular anatomical reasons. In young children, the metaphysis of the vertebral body exhibits a rich vascular ring, which creates an anastomosis with the adjacent metaphyseal vascular ring through several branches adjacent to the posterolateral region of the disk. The vascular supply of the disc in children is made up of vessels that come across the cartilaginous vertebral plate and into the disc ring. After 8 years of age, these vessels disappear, but a dense anastomotic network of vessels remains in the postero-lateral region of the disc. Thus, some authors consider that all bacterial infections are primarily located in the metaphyseal region of the vertebral body, with the microorganism first crossing the cartilaginous vertebral plate, running through the surface of the disc via the anastomotic branches, infecting the adjacent vertebral metaphysis and finally reaching the disc space between the two vertebral bodies involved. What is opinion about this physiopathological explanation?’
R2: We agree with this point, and feel that the comment reflects ongoing uncertainty regarding the pathophysiology and anatomical changes underlying spondylodiscitis. However, the distinction between pure discitis, and spondylodiscitis (i.e. with involvement of the adjacent vertebra) is not one that is clinically meaningful, since the approach to treatment is the same. As described above (R1) we have removed the term discitis from the opening sentence of the introduction to remove any confusion around diagnostic labels with imprecise and overlapping definitions.
C3: ‘Inclusion and exclusion criteria have to be better explained.’
R3: In response to this comment the methods section of the manuscript has been revised to include an explicit statement of the inclusion and exclusion criteria;
Lines 72-77: ‘The inclusion criteria were; adult patients (aged 16 years or over), a radiological diagnosis of pyogenic spondylodiscitis and a bacterial aetiology confirmed by microbiological culture or a bacterial aetiology not confirmed but thought likely by the treating infectious diseases team. The exclusion criteria were, children (aged under 16 years), confirmed tuberculous (n=2) or fungal (n=1) spondylodiscitis or the absence of clinical outcome data (n=7).’
C4: ‘In the series, the authors collected 20 patients that presented surgical site infections following spinal surgery. I do not think that these cases may be considered as pyogenic spondylodiscitis, but they are secondary spinal infections. To my opinion, surgical site infections following spinal surgery are affections, with different microbiological reality, treatment and outcome.’
R4: We accept that post-surgical infections may have important microbiological differences compared to spondylodiscitis arising from other causes (e.g due to haematogenous spread or spread from adjacent structures), however the clinical and radiological manifestations of spondylodiscitis in this context, and treatment provided, are indistinguishable. The data in this study bear this point out. We did observe clear differences in the infecting pathogens, as illustrated by the significant association between post-surgical infections and the isolation of coagulase-negative Staphylococci (lines 147-149). Importantly however, the post-surgical infection aetiology did not predict clinical outcome (see Table 2) and there was no significant difference in either duration of antibiotic therapy (mean antibiotic duration was 137 days in post-surgical patients vs. 108 days in other patients, p=0.194) or surgical treatment rates (surgical intervention in 6/20 post-surgical patients vs. 32/71 other patients, p=0.227). Furthermore, there is clear precedent in the literature for the inclusion of these post-surgical cases in such series. For example, previous observational and randomised controlled trials on spondylodiscitis (e.g. the RCT by Bernard et al) included post-surgical infections in their analyses [1-4]. In view of this we have not removed these cases from our study and instead have acknowledged this issue in the limitations section of our discussion in the revised manuscript:
Line 302-307: ‘Finally, it can also be argued that post-surgical infections may represent a distinct pathogenic process to spondylodiscitis from other causes, particularly in view of the differences in microbiological aetiology reported here. However, including post-surgical cases reflects the reality that the treatment principles are similar and is consistent with the approach taken in other cases series and the randomised controlled trial by Bernard et al. [7, 8, 11, 14].’
C5: ‘If you consider that these cases of surgical site infections following spinal surgery have to be collected in your series, it will be important to know what surgeries were realized (discectomies, spondylodesis, radicular neurolysis for radiculopathies, decompressive laminectomies,…?).
R5: In order to include these additional data, the following revision has been made to the manuscript:
Lines 104-107: ‘In addition, 20 (22%) patients had a history of spinal surgery within the past 3 years; 6 infections occurred after decompressive laminectomy, 6 after discectomy, 6 after spinal fixation, 1 after vertebroplasty, 1 after insertion of a spinal stimulator and 1 after resection of a retroperitoneal sarcoma.’
We have also noticed that in Figure 2D there was an error whereby the post-surgical and unknown sources of infection were incorrectly labelled. We apologise for this error which has now been corrected in the revised figure.
C6: ‘More than 50% of their patients had impairment of motor function. Could authors explained the real cause of neurological impairment (compression due to epidural abscess, instability, narrow canal,…?)’
R6: In order to include these additional data, the following revision has been made to the manuscript:
Lines 112-117: ‘At presentation 46 patients (51%) had a Frankel grading score of A to D indicating various degrees of impairment of motor function; of these patients 31 had a spinal epidural abscess, 6 had vertebral instability, 5 had a prevertebral or paravertebral abscess and in the remaining 4 spondylodiscitis was the only identifiable cause for the neurological impairment, which was presumably secondary to nerve root impingement.’
C7: ‘Could the authors explain which were their indications for performing drainage of epidural abscesses?’
R7: This is an excellent question. However, it is difficult to comment on a general approach to the surgical management of epidural abscesses as this study describes management in a heterogenous population by a number of different clinicians. Furthermore, the reasoning underlying specific treatment approaches was rarely explicitly described in clinical documentation. In general, the decision on whether to operate was based on the radiological appearance of the abscess (particularly abscess size and the presence of spinal cord/cauda equina compression), the resultant clinical features (particularly the presence of neurological deficits) and the operative risk due to the performance status and co-morbidities of the individual patient. The uncertainty regarding the role of surgical intervention in pyogenic spondylodiscitis, including in the presence of spinal epidural abscess, is acknowledged in lines 282-293 of the discussion.
C8: ‘In the discussion, the authors evoke the severity of illness index; could they explain index illness to the readers?’
R8: In the discussion we were not referring to the severity of illness index. This comment therefore helpfully identifies our use of a phrase (“index illness”, meaning presenting illness) that is open to misinterpretation. In order to avoid any confusion, the following revision has been made to the manuscript:
Lines 216-217: ‘As we identified no evidence of infection relapse, these sequelae are most likely to relate to the damage done during the initial illness.’
References
1. Bernard, L., et al., Antibiotic treatment for 6 weeks versus 12 weeks in patients with pyogenic vertebral osteomyelitis: an open-label, non-inferiority, randomised, controlled trial. Lancet, 2015. 385(9971): p. 875-82.
2. D'Agostino, C., et al., A seven-year prospective study on spondylodiscitis: epidemiological and microbiological features. Infection, 2010. 38(2): p. 102-7.
3. Garkowski, A., et al., Infectious spondylodiscitis - a case series analysis. Adv Med Sci, 2014. 59(1): p. 57-60.
4. McHenry, M.C., K.A. Easley, and G.A. Locker, Vertebral Osteomyelitis: Long-Term Outcome for 253 Patients from 7 Cleveland-Area Hospitals. Clinical Infectious Diseases, 2002. 34(10): p. 1342-1350.
Reviewer 2 Report
1. Abstract line 20- do not begin sentence with figures
2. Define unfavorable outcome in abstract
3. What were the risk factors identified in the study, include in abstract
Intoduction
1. Line 63-67 are conclusions and should be removed from introduction
Methods
2. Describe a bit about the hospital. What kind of servises are offered and the total patient load during the study period.
3. Was the study prospective? According to my assessment this is a retrospective study. Please change and see why below
4. How many cases of TB or fungal discitis were found?
5. “The degree of neurological impairment was retrospectively determined using the Frankel grading scale of neurological injury [5] using data collected at the time of clinical assessment (Table 1).”
Why was the neurological assessment done retrospectively when the study was prospective? If it was done in the way it is currently described it is a retrospective study. Please change in methods section.
My point is substabtiated further in line 83 and in the results line 96
“Outcome data were collected retrospectively using clinical and electronic patient records.”
Results
6. Line 96-Full outcome data were available on 91/98 patients.- Why was information missing if the study prospective.
7. Please put number or percentage in the Figure 2 (A to D), to make it more visual friendly.
8. Please provide antibiotic sensitivity data for the cultured microorganisms. This will increase the value of the article tremendously.
9. A multivariate logistic regression needs to done for the variable included in table 3 for identification of risk factors for unfavourable outcome.
10.Please put number or percentage in the Figure 3 and Figure 4
11.Please write a conclusion section
Author Response
Department of Infection and Tropical Medicine,
Royal Victoria Infirmary,
Queen Victoria Road, Newcastle upon Tyne,
NE1 4LP,
United Kingdom
17th October 2018
Re: medsci-363764 - Pyogenic spondylodiscitis: risk factors for adverse clinical outcome
Dear Reviewer
Thank you for your email of 10th October 2018 giving us the opportunity to respond to the comments made following the review of our article. We have responded to each of these comments below and made requested changes to the manuscript. We believe the manuscript is improved as a result of these changes and hope that it will now be acceptable for publication in Medical Sciences.
In the text below we have listed the comments in the order that they appeared in each review email and numbered them C1, C2 etc. with our corresponding responses beneath the relevant comment and numbered R1, R2 etc. As requested we have produced a revised manuscript in which our changes are clearly marked. The line numbers included in the responses corresponds to those in the ‘Simple Markup’ tracked version of the Word Document.
We would be happy to provide any further information that you require.
Yours faithfully
Dr John Widdrington, on behalf of the authors
Open Review
(x) I would not like to sign my review report
( ) I would like to sign my review report
English language and style
( ) Extensive editing of English language and style required
( ) Moderate English changes required
(x) English language and style are fine/minor spell check required
( ) I don't feel qualified to judge about the English language and style
Yes | Can be improved | Must be improved | Not applicable | |
Does the introduction provide sufficient background and include all relevant references? | ( ) | (x) | ( ) | ( ) |
Is the research design appropriate? | ( ) | ( ) | (x) | ( ) |
Are the methods adequately described? | ( ) | ( ) | (x) | ( ) |
Are the results clearly presented? | ( ) | ( ) | (x) | ( ) |
Are the conclusions supported by the results? | ( ) | ( ) | (x) | ( ) |
Comments and Suggestions for Authors
C1: ‘Abstract line 20- do not begin sentence with figures’
R1: The abstract has been revised in order to respond to comments C1, C2 and C3 while keeping to the 200 word limit. The revised abstract is shown below:
Lines 20-34: ‘We aimed to describe the clinical features and outcomes of pyogenic spondylodiscitis and to identify factors associated with an unfavourable clinical outcome (defined as death, permanent disability, spinal instability or persistent pain). In our tertiary centre 91 cases were identified prospectively and a retrospective descriptive analysis of clinical records was performed prior to binary regression analysis of factors associated with an unfavourable outcome. A median 26 days elapsed from the onset of symptoms to diagnosis and 51% of patients had neurological impairment at presentation. A microbiological diagnosis was reached in 81%, with Staphylococcus aureus most commonly isolated. Treatment involved prolonged hospitalisation (median stay 40.5 days), long courses of antibiotics (>6 weeks in 98%) and surgery in 42%. While this was successful in eradicating infection, only 32% of patients had a favourable clinical outcome and six patients (7%) died. Diabetes mellitus, clinical evidence of neurological impairment at presentation, a longer duration of symptoms and radiological evidence of spinal cord or cauda equina compression were independent factors associated with an unfavourable outcome. Our data indicates that spondylodiscitis is associated with significant morbidity and suggests that adverse outcomes may be predicted to an extent by factors present at the time of diagnosis.’
C2: ‘Define unfavorable outcome in abstract’
R2: The following statement is included in the revised abstract:
Lines 21-22: ‘…and to identify factors associated with an unfavourable clinical outcome (defined as death, permanent disability, spinal instability or persistent pain).’
C3:’What were the risk factors identified in the study, include in abstract’
R3: The following statement is included in the revised abstract:
Lines 30-32: ‘Diabetes mellitus, clinical evidence of neurological impairment at presentation, a longer duration of symptoms and radiological evidence of spinal cord or cauda equina compression were independent factors associated with an unfavourable outcome.’
C4: ‘Introduction Line 63-67 are conclusions and should be removed from introduction’
R4: As the reviewer has suggested, this section has been removed from the introduction. The information that was included here has been re-worded to form part of a conclusion section. Please see the response to C14 for more details about the conclusion section.
C5: ‘Methods. Describe a bit about the hospital. What kind of servises are offered and the total patient load during the study period.’
R5: In order to clarify this the following revision has been made to the manuscript:
Lines 68-71: ‘Data were collected as part of routine clinical service evaluation at Newcastle Upon Tyne Hospitals NHS Trust, a large teaching trust with 1,800 beds that manages over 1.72 million patient contacts per year and is a tertiary referral centre for both Spinal Surgery and Infectious Diseases services.’
C6: ‘Was the study prospective? According to my assessment this is a retrospective study. Please change and see why below’
R6: Upon reflection in response to C6, C8 and C9, we understand that our description of the study design in our original manuscript was potentially misleading and apologise regarding the lack of clarity. To be clear, in our study consecutive cases of pyogenic spondylodiscitis were identified prospectively at the point of referral to the Infectious Diseases department. Subsequently the clinical and outcome data were retrospectively collected and analysed.
To avoid any misconceptions, the study design explicit the following changes have been made to the revised manuscript:
Abstract - lines 22-23: ‘In our tertiary centre 91 cases were identified prospectively and a retrospective descriptive analysis of clinical records was performed…’
Methods - lines 77-81: ‘Subsequent to the prospective case identification, anonymised details of clinical and radiological presenting features, microbiological diagnosis, surgical and antimicrobial treatment were retrospectively obtained from clinical and laboratory records and recorded using a standardised proforma and/or directly entered onto a database (Microsoft Excel, Redmond, WA, USA).’
Discussion - lines 294-295: ‘Whilst cases were identified prospectively, outcome data were collected and analysed retrospectively and therefore may be subject to bias.’
C7: ‘How many cases of TB or fungal discitis were found?’
R7: In order to include these additional data, the following revision has been made to the manuscript:
Lines 75-76: ‘The exclusion criteria were…confirmed tuberculous (n=2) or fungal (n=1) spondylodiscitis…’
C8: ‘The degree of neurological impairment was retrospectively determined using the Frankel grading scale of neurological injury [5] using data collected at the time of clinical assessment (Table 1).
Why was the neurological assessment done retrospectively when the study was prospective? If it was done in the way it is currently described it is a retrospective study. Please change in methods section.
My point is substabtiated further in line 83 and in the results line 96
“Outcome data were collected retrospectively using clinical and electronic patient records.’
R8: The data collection and analysis was carried out retrospectively. Please see the response to C6 regarding the changes made to the manuscript to clarify this and ensure the study design is explicit.
C9: ‘Results Line 96-Full outcome data were available on 91/98 patients. Why was information missing if the study prospective?’
R9: As discussed previously, while the case identification was prospective the study data was collected retrospectively. Seven patients were lost to follow-up due to transfer to a hospital outside of the region during treatment or a failure to attend out-patient appointments. In these patients the final clinical outcome was unclear and for this reason they were not included in the final analysis. This is highlighted in the statement of the exclusion criteria included in the revised manuscript:
Lines 75-77: ‘The exclusion criteria were… or the absence of clinical outcome data (n=7).’
C10: ‘Please put number or percentage in the Figure 2 (A to D), to make it more visual friendly.’
R10: As the reviewer has suggested Figure 2 has been revised in order to include exact numbers for the data displayed in the bar charts.
C11: ‘Please provide antibiotic sensitivity data for the cultured microorganisms. This will increase the value of the article tremendously.’
R11: Antimicrobial resistance is a complex and fascinating topic, but we feel that including a systematic analysis of this is beyond the scope of our current study. In particular, we feel that these data would be challenging to present in a concise and meaningful manner given the range of organisms that were cultured. Furthermore, antimicrobial resistance was not a major issue in the patients included in our study, for example only 1/35 (3%) of patients in which Staphylococcus aureus was isolated had MRSA, and in all cases where a causative organism was identified, antibiotic therapy was targeted to the sensitivity profile.
In order to acknowledge this point, the following revision has been made to our manuscript:
Lines 144-146: ‘Although antimicrobial resistance data is not included here, this was not a major problem in our cohort of patients, as illustrated by the finding that only 1/35 (3%) of patients in which Staphylococcus aureus was isolated had a methicillin-resistant strain.’
C12: ‘A multivariate logistic regression needs to done for the variable included in table 3 for identification of risk factors for unfavourable outcome.’
R12: A binary logistic regression analysis was carried out in the analysis presented in Table 3 as the observed dependent variable (clinical outcome) has only two possible outcomes (favourable or unfavourable). A multivariate logistic regression is used when there are three or more possible outcomes for the dependent variable. This point has helpfully highlighted an error in our abstract; the term ‘binary regression analysis’ replaces ‘multivariate regression analysis’ in the revised version (see response to C1).
C13: ‘Please put number or percentage in the Figure 3 and Figure 4’
R13: As the reviewer has suggested Figure 3 and 4 have been revised in order to include exact numbers or percentages for the data displayed in the bar charts.
C14: ’Please write a conclusion section’
R14: In response to this comment the manuscript has been revised to include the following conclusion section:
Lines 309-318: ‘Spondylodiscitis is a rare but serious condition that leads to significant long-term morbidity. In this study we have identified several important findings that contribute to our understanding of this neglected disease. These include the observations that: a) there is often a substantial interval between symptom onset and diagnosis, b) long-term sequelae such as pain are common despite adequate treatment of infection, and c) several easily identifiable baseline factors (including duration of symptoms, diabetes mellitus and neurological involvement at presentation) appear to predict adverse clinical outcome. Our findings highlight the need for further research, particularly to assess the impact of interventions to reduce diagnostic delay and to optimise antibiotic and/or surgical management. Such research has considerable potential to improve the outcome for patients with spinal infections.’
Round 2
Reviewer 2 Report
The authors have done most of the changes requested by me, except Figure 4, which still requires numbers to be put in front of the bar chart.